# MEMS Enabled Miniature Two-Photon Microscopy for Biomedical Imaging

**DOI:** 10.3390/mi14020470

**Published:** 2023-02-17

**Authors:** Xiaomin Yu, Liang Zhou, Tingxiang Qi, Hui Zhao, Huikai Xie

**Affiliations:** 1Key Laboratory of Biological Effect of Physical Field and Instrument, Department of Electrical and Electronic Engineering, Chengdu University of Information Technology, Chengdu 610225, China; 2Department of Electrical and Computer Engineering, University of Florida, Gainesville, FL 32611, USA; 3BIT Chongqing Institute of Microelectronics and Microsystems, Chongqing 401332, China; 4Foshan Lightview Technology Co., Ltd., Foshan 528000, China; 5School of Integrated Circuits and Electronics, Beijing Institute of Technology, Beijing 100081, China

**Keywords:** miniaturized two-photon microscopy, head-mounted, microendoscopy, MEMS mirrors, axial scanning

## Abstract

Over the last decade, two-photon microscopy (TPM) has been the technique of choice for in vivo noninvasive optical brain imaging for neuroscientific study or intra-vital microendoscopic imaging for clinical diagnosis or surgical guidance because of its intrinsic capability of optical sectioning for imaging deeply below the tissue surface with sub-cellular resolution. However, most of these research activities and clinical applications are constrained by the bulky size of traditional TMP systems. An attractive solution is to develop miniaturized TPMs, but this is challenged by the difficulty of the integration of dynamically scanning optical and mechanical components into a small space. Fortunately, microelectromechanical systems (MEMS) technology, together with other emerging micro-optics techniques, has offered promising opportunities in enabling miniaturized TPMs. In this paper, the latest advancements in both lateral scan and axial scan techniques and the progress of miniaturized TPM imaging will be reviewed in detail. Miniature TPM probes with lateral 2D scanning mechanisms, including electrostatic, electromagnetic, and electrothermal actuation, are reviewed. Miniature TPM probes with axial scanning mechanisms, such as MEMS microlenses, remote-focus, liquid lenses, and deformable MEMS mirrors, are also reviewed.

## 1. Introduction

Two-photon microscopy (TPM) was first demonstrated by W. Webb et al. in 1990 [1] and was based on nonlinear two-photon excitation (TPE) [2]. TPE is an extremely rare phenomenon in nature that requires two photons interacting with the same molecule simultaneously (~10^−16^ s) [3]. As the probability of TPE depends on the light intensity quadratically, focused high-intensity pulsed femtosecond lasers must be used. The TPE-induced fluorescence is thus confined only within a small focal volume, which provides the TPM with the ability of optical sectioning without the need for extra components, such as pin holes in confocal microscopy.

Two-photon microscopy falls into the category of laser scan microscopy, where the focal point of the excitation laser quickly scans across the sample. Thus, emitted fluorescence is collected point by point across the field of view (FOV) and each individual image pixel is reconstructed from the signal acquired from a known point in the sample. With the development of tunable femtosecond pulsed lasers [4], TPM has been widely implemented in research labs. A simplified schematic diagram of a TPM is shown in Figure 1. A femtosecond pulsed laser beam is shone into an intensity controller to adjust the power intensity and then deflected by two in-plane transversal scanners. Then, the laser beam is relayed into a dichroic mirror via a scan lens and a tube lens, where the beam size is also expanded. After this, the excitation laser beam is selectively folded toward a high-numerical-aperture (NA) objective lens and sharply focused on a region of interest in a sample, e.g., the head of a mouse. Thus, the focused laser beam excites fluorescence out of the sample, which is then backscattered and passes through the objective lens and the dichroic mirror, and finally is focused and selectively detected by a photodetector, such as a photomultiplier tube (PMT). For the detailed design and setup of a TPM, please refer to [5,6,7].

Due to its intrinsic capacity for optical sectioning, deep tissue penetration, and low photobleaching and photodamage in a small volume of the focal plane, TPM has been widely used in biological imaging, especially for in vivo noninvasive optical brain imaging [3] or laser scanning microendoscopic imaging [8]. With the help of genetically encoded calcium indicators (GECIs) [9,10], TPM has revolutionized the field of calcium imaging in living animals [11,12,13]. For example, TPM has been used to study neuronal activity [14,15,16], neuronal plasticity [17,18], and neurodegenerative diseases [19,20]. To obtain stable in vivo brain imaging and study the neuron activities in a relatively large depth, head fixation is deployed for a conventional bench-top TPM in which the head of a small animal is constrained under the two-photon microscopy objective [16]. Thus, either anesthesia or extensive training is needed for the constrained animals, which limits the investigation and understanding of brain activities in a real physiological environment. For in vivo microendoscopic imaging, the envelope of the TPM scanner should reside within an endoscope port with a diameter of 7 mm [21]. Moreover, the scanning process should be fast such that physiological motion, such as the breathing or heartbeat of the subject, does not compromise the fidelity of the image.

To overcome these challenges, a miniature TPM (mTPM) imaging probe head-mounted on a freely behaving animal or inserted into an endoscope must be applied, as illustrated in Figure 2. Optical fiber permits the delivery of femtosecond pulses from remote bulky laser sources and enables the flexibility of the imaging system. Microelectromechanical systems (MEMS) technologies make it possible to achieve miniaturized two-photon microscopy as MEMS devices are small and fast. MEMS-enabled miniature TPM has been exploited extensively in the last decade, and significant progress has been achieved [4]. However, there is still no comprehensive review found in the literature.

In this paper, we will review the latest advancements in miniature TPMs based on MEMS technologies. The performance of mTPM is often limited by the integration of miniaturized optical components, architecture, and MEMS microscanners based on various actuation mechanisms into a small probe or headpiece. Hence, the crucial miniaturized components in a miniature TPM are firstly discussed in Section 2; then, various MEMS actuator mechanisms for lateral scanning and other techniques for axial scanning are reviewed in Section 3 and Section 4; and, finally, Section 5 summarizes the challenges and outlook of mTPM scanners for in vivo biomedical imaging applications.

## 2. Miniature Two-Photon Microscopy

Miniature TPM probes can be used for scanning with two-photon microendoscopes for in vivo tissue imaging or mounted on top of live animals for long-term studies of neuronal activities while observing animal behaviors concurrently. To achieve in vivo TPM imaging, all subsystems must be miniaturized. These crucial components and basic building blocks for the miniaturization of TPMs, including fiber optics, objective lenses, and laser scanners, will be reviewed first.

### 2.1. Fiber Optics

Considering the weak two-photon excitation, femtosecond pulsed laser sources with high instant power intensity are usually used. The ultrashort femtosecond laser source usually has a pulse width of approximately 140 fs and contains a spectral bandwidth of over 100 nm. At present, no miniature high-power femtosecond laser sources are available. Therefore, optical fibers are used to circumvent the dilemma as a replacement for conventional bulk and free-space optics in a bench-top TPM.

There are two practical approaches to miniaturizing the fiber-coupled TPM probe design. One is a two-fiber configuration. A standard single-mode fiber (SMF) [22] or a hollow-core photonic bandgap fiber (PBF) [23] is used for femtosecond pulse delivery. The excitation efficiency of TPM signals is affected by the dispersion and pulse broadening of laser pulses. Femtosecond pulses suffer severe chromatic dispersion when propagating through SMA and can become broadened to picosecond pulses, which would deteriorate the excitation efficiency of fluorescence. The hollow-core PBF exhibits ~20 times lower dispersion than SMF because most light propagates in the air core. However, signal attenuation can be severe in the hollow-core PBF because the spectrum bandwidth is reduced due to its narrow low-loss propagation window [24]. The collection efficiency of the excited TPM signals is affected by the NA of the fibers. To improve the coupling efficiency of emission fluorescence, another multi-mode fiber with a large core diameter and high NA is used, as shown in Figure 3a. Plastic optical fibers with diameters ranging from 0.4–2 mm and a high NA [25,26] are used in miniature TPM probes, but the flexibility may be limited due to the stiffness incurred by the large diameter. An alternative method is to replace the large core fiber with a fiber bundle. For example, 800 glass optical fibers fused into a diameter of 1.5 mm at both ends are used to collect fluorescence [27]. In a two-fiber configuration, the large size of the TMP probe is not suitable for endoscopic imaging due to the two fibers. Therefore, a single-fiber configuration is used to guide the excitation and emission beams in a mTPM probe, as shown in Figure 3b.

**Figure 2 micromachines-14-00470-f002:**
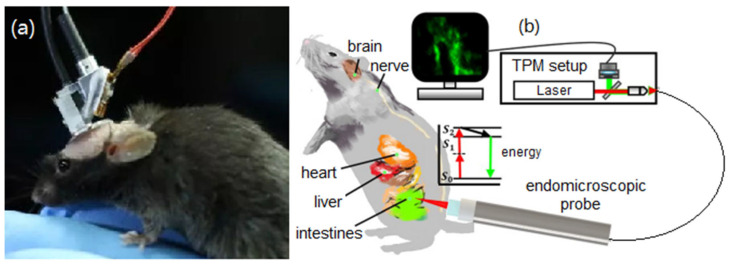
Concept of a miniature TPM. (**a**) Miniature TPM headpiece mounted on top of a small animal. Reproduced with permission from [27]. Copyright 2017, Spring Nature. (**b**) TPM probe for endoscopic imaging. Reproduced with permission from [8]; Copyright 2019, Springer Nature.

In a single-fiber configuration, the utilized fiber should offer high efficiency for both femtosecond pulse delivery and fluorescence collection. An air-core photonic crystal fiber (PCF) and double-clad photonic crystal fiber (DC-PCF) have been demonstrated in ultrashort femtosecond pulse delivery. An air-core fiber allows the delivery of high-peak-intensity femtosecond pulses with a ~65% coupling efficiency [26], as shown in Figure 4a. DC-PCF has a solid core that operates as an SMF and an air-hole structured inner cladding that functions as a multimode fiber, as shown in Figure 4b, which have been employed for the delivery of ultrashort pulses via the single-mode core and the collection of the two-photon fluorescent light via the inner cladding layer with a large diameter and high NA [28,29]. The advantages of a single-fiber configuration based on a DC-PCF not only include the possibility for a smaller TPM probe but also make the detection efficiency of a nonlinear optical microscopy system approximately two orders of magnitude higher than that of the single-mode fiber-based microscope, mainly due to the large core size and high NA of the inner cladding [30].

### 2.2. Objective Lenses

High-performance objective lenses with high NA in a bench-top TPM are large and heavy, and they have been widely used for ex vivo or intra-vital imaging in biomedical research laboratories. However, they cannot be integrated into miniature TPM probes, which require miniaturized optics for optical relay and focusing. Gradient refractive index (GRIN) lenses have a spatially varying refractive index and exhibit distinct optical properties due to their cylindrical shape, small diameter, and micro-scale resolution. Compound gradient refractive index (GRIN) lenses, which were first used as minimally invasive optical relays for convectional TPM in 2004 [31], were thus gradually used to couple with miniature TPM probes. A high-performance 1-mm-diameter achromic miniature GRIN objective was used by Liang et al. [32] for in vivo two-photon imaging in 2017. GRIN lenses can make imaging probes compact, but they have a limited FOV and a short working distance.

Some other TPM probes use miniature spherical or aspherical lenses as objective lenses [24,33,34]. Miniature compound objectives have better optical performance. For example, objectives integrated with multiple GRIN lenses and aspherical lenses exhibit sub-micron resolution but at the price of a limited FOV and short working distance [27]. Together with the balance of a wide FOV, a long working distance, and micron resolution, a custom TPM probe with lower NA objectives was demonstrated to allow long-term recurrent imaging mounted on the head of a freely behaving mouse [35]. In addition, Chung et al. reported a Blu-ray disk lens as the objective for a compact TPM head [36]. This lens was an aspheric lens with 0.85 NA, 2.5 mm in diameter, and 1.5 mm in thickness, composed of plastic. By combining the Blu-ray disk with a commercially available spherical tube lens pair, the scanner achieved 0.6 μm lateral resolution but only a FOV of 178 × 280 μm^2^.

### 2.3. Transverse Scan Mechanisms

For a single-laser-beam-based TPM, a scanning mechanism must be employed to achieve transverse one-axis or two-axis optical scanning for two-photon imaging. Fiber scanning and MEMS-based scanning are the two major transverse scanning mechanisms. A fiber scanner includes proximal and distal scanning. Galvanometric mirrors are powerful and can work with fiber bundles to provide proximal scanning, i.e., scanning the proximal end of a TPM probe [37], as shown in Figure 5a. The advantage of this scan configuration is that the size of the TPM probe is not related to the scanner size due to the spatial separation. However, this scanning configuration incurs suboptimal optical delivery and collection efficiency due to the unavoidable gaps between the individual fibers in the fiber bundle. Thus, it is not optimal for the miniaturization of TPM probes. Distal scanning, located near the sample, is thus favored in fiber optic imaging modalities.

Fiber tip scanning, as illustrated in Figure 5b, is one popular transverse scan mechanism enabling distal scanning. Fiber tip scanning has been widely used in various imaging modalities [32,38,39,40,41]. To obtain a more compact scanner for microendoscopy applications, piezoelectric [32,42,43,44] excited actuators have been widely used to drive the mechanical resonance vibration of the fiber tips. In 2014, Do et al. [42] developed a nonresonant fiber-optic raster scanning two-photon endomicroscope using a quarter-tubular piezoelectric (PZT) actuator with an outer diameter of 3.5 mm and rigid distal length of 30 mm. Its scanning speed and scanning area can be controlled by adjusting the applied voltages of the quarter tubular PZT actuator. Two-photon images of a stained mouse kidney section were observed with an image acquisition time of 1.6 s and a pixel size of 256 × 256. In 2015, Ducourthial et al. [44] presented a miniaturized fiber scanner of 2.2 mm outer diameter, as illustrated in Figure 6a, which allowed two-photon excited autofluorescence (TPEF) imaging at 8 frames per second with a FOV of 450 μm for 60 V on the PZT. Label-free in vivo microendoscopy images of the kidney of an anesthetized mouse were acquired with transverse and axial resolutions of 0.8 μm and 12 μm, respectively, a FOV as large as 450 × 450 μm^2^, and a working distance of 660 μm in water. Aiming for label-free functional histology in vivo, a more compact and flexible PZT-based fiber scanner with a smaller outer diameter of 2 mm and a rigid length of 35 mm, as shown in Figure 6b, was reported by Liang et al. in 2017 [32]. The resonance of the fiber tip was 1.4 kHz for a spiral scanning pattern, and the spatial resolution of the microendoscope was ~0.7 μm laterally and ~6.5 μm axially. This two-photon microendoscopy system successfully achieved label-free visualization of the mucosa of the small intestine of a mouse in vivo (Figure 6d,e). The advantages of a PZT tube-based fiber scanner include the low cost, low power consumption, high resolution, and the small outer diameter of the distal end, which could potentially be fabricated to be less than 1.5 mm. However, tubular PZT-actuated scanning methods cause significant nonuniformity between the inner and outer parts of the scanning area. High-driven applied voltages to the PZT actuators are also a safe relevant concern for microendoscopic imaging.

At the same time, microelectromechanical systems (MEMS) scanning technologies, which are uniquely suited to the combination of actuators to guide light in very small volumes with micro-optical elements for in vivo beam scanning and imaging, have emerged as a versatile miniature transverse scanning mechanism, as illustrated in Figure 5c. Large force actuators as required by fiber tip scanning are not necessary for driving MEMS mirrors. Due to their small size, high-performance scanning, and mass production ability, MEMS mirrors are becoming one of the best types of scanners enabling miniature optical scan probes. Therefore, the following section will focus on MEMS-based miniature TPMs in detail.

## 3. MEMS-Based TPM

With a small footprint, low power consumption, relatively high scan speed, low cost, and ease of integration, MEMS mirrors have been successfully incorporated into various optical systems, such as displays [45], optical switching [46], structured illumination [47], LiDAR [48], optical coherence tomography (OCT) microendoscopy [49], and confocal microendoscopy [50]. The application of MEMS mirrors to miniature nonlinear optical endoscopic probes was first reported by Fu et al. in 2006 [51]. Following this, tremendous progress has been made in the miniaturization of TPM probes, along with the development of various MEMS and relative micro-optics. Various actuation principles, such as electrostatic, electromagnetic, and electrothermal principles, can be employed to obtain MEMS mirrors. Aided by MEMS mirrors, bench-top TPM systems have been evolving into miniature probe-based TPM systems, which greatly expand their application range, especially for in vivo two-photon imaging, such as for live, freely moving animals.

### 3.1. Electrostatic MEMS-Based TPM

The electrostatic force generated from two counter electrodes can be widely used to drive MEMS mirrors. The first MEMS mirror, developed by Peterson in 1980, was actuated by a parallel plate structure [52], as shown in Figure 7a. This parallel structure has been widely used to deflect mirror plates with small apertures and is especially known for driving small mirror arrays, such as in Digital Micromirror Devices (DMD) [53,54]. The disadvantage of the parallel plate actuator is its scan range and high driving voltage due to the large separation gap. A comb drive structure is developed for more efficient driving to reduce the pull-in effect observed in the parallel plate structure. The comb driver was first introduced by Tang et al. in 1990 [55]. Being convenient to generate torques, the first vertical (out-of-plane) comb driver was presented by Selvakumar et al. in 2003 [56], as shown in Figure 7b. In addition, angular or curled comb drives were also developed to deflect mirror plates [57,58,59,60], as illustrated in Figure 7c,d. For example, vertical-comb-drive MEMS mirrors have been used to develop miniature confocal microscopy [50,61]. Planar-comb-drive-based MEMS mirrors have also been developed by using nonlinear forces [62,63].

A series of TPM probes based on electrostatic MEMS mirrors have been reported. Piyawattanametha et al. demonstrated the first two-photon microscopic probe prototype based on an electrostatic MEMS mirror in 2006 [64]. The two-axis MEMS mirror had a gimbal design (Figure 8a) with two torsional springs (Figure 8b) and two sets of vertical comb actuators (Figure 8c). The central mirror was 750 × 750 µm^2^, and the total die size was 3.2 × 3 mm^2^. The resonant frequencies of the MEMS mirror were 1.76 kHz and 1.02 kHz at the inner and outer axes, respectively, and the maximum optical scan angles were ±7.6° and ±3.0° for the inner and outer axes at 80 V and 160 V, respectively. Two-photon images of pollen grains were acquired using this TPM probe, with a field of view (FOV) of 250 × 90 µm^2^ (Figure 8d). After this, the same group optimized the miniature TPM probe design with a weight of only 2.9 g and a smaller size of 20 × 19 × 11 mm^3^ (Figure 8e,f) [23]. The optical scan angles were around ±5° and ±4.3° in dc operation, and the resonant frequencies were 1.08 kHz and 0.56 kHz for its inner and outer axes, respectively. The probe yielded a lateral and axial resolution of 1.29 ± 0.05 μm and 10.3 ± 0.3 μm, respectively, and the maximum FOV was 295 × 100 μm^2^. This MEMS-based microscopic probe was demonstrated by imaging the neocortical microvasculature in anesthetized adult mice at four frames per second, as shown in Figure 8g. In addition, a depth scan of 270 μm was obtained using a dc micromotor. The optical design based on small GRIN lenses made the probe compact, but its FOV and depth of focus were limited.

In 2009, Tang et al. demonstrated endoscopic multiphoton microscopy (MPM) probes using electrostatic MEMS mirrors [24,33] with a single DC-PCF configuration. The MEMS mirror was gimbal-less, which was first reported by Milanovic et al. [65] in which the vertical comb driving the actuators and beams occupied most of the die area, resulting in a very low fill factor. The drawback could be circumvented by bonding a separate large mirror plate, where a 2-mm-diameter mirror was bonded on a 3.3 × 2.6 mm^2^ die. The resonance frequencies of this MEMS mirror were 1.26 kHz and 780 Hz at its two axes, respectively. The maximum optical scan angle was around 20°, and the maximum number of resolvable focal spots was 720 × 720. The diameter of the probe was 10 mm, which could be potentially reduced by using smaller lenses and improved machining for clinical microendoscopic applications. Duan et al. demonstrated another multiphoton microendoscope in 2015 [25]. This probe was based on a two-fiber configuration, as shown in Figure 9a. This gimbal MEMS mirror had a diameter of 1.8 mm and a footprint of 3 × 3 mm^2^ (Figure 9d). The resonance frequencies in the X and Y axes were 2.91 kHz and 0.805 kHz, respectively. The maximum scan angle of the MEMS mirror was ±4.5° at 40 V_pp_ with twice the resonance frequencies at both axes. This probe thus formed a FOV of 300 × 300 μm^2^ and achieved lateral and axial resolutions of 2.03 and 9.02 μm, respectively. Using this 3.4-mm-diameter and 26-mm-long imaging probe (Figure 9b,c), two-photon microendoscopic images of the individual cell nuclei of a mouse colon in vivo at 5 frames per second were successfully observed, as shown in Figure 9e,f.

In 2017, Zong et al. reported a head-fixed TPM probe with a two-fiber configuration that weighed only 2.15 g and occupied 1 cm^3^, based on an electrostatic MEMS mirror [27], as shown in Figure 10a. The MEMS mirror was 0.8 mm in diameter and packaged into a footprint of 9 × 9 mm^2^. Its resonant frequency was 6 kHz, and its maximal optical scan angle reached ±10° at 10 V. It achieved a lateral resolution of 0.6 μm and an axial resolution of 3.4 μm. This MEMS probe provided an imaging rate of 40 fps with a frame size of 256 pixels × 256 pixels at a FOV of 130 × 130 μm^2^. While the size of the probe limited its application for two-photon microendoscopic imaging, it was successfully used for the imaging of neuronal activities at the level of the spines of freely moving mice, and it demonstrated almost the same performance as that of a bench-top system. The miniature version of this TPM probe was limited by the large package size of the MEMS mirror (Figure 10b). Other drawbacks of the probe include the small FOV, the short working distance (~170 μm), and the lack of an axial scanning mechanism. In 2021, the same team further optimized the design of the probe to obtain a size of 16 × 9 × 30 mm^3^ and a weight of 4.2 g [35], as shown in Figure 10c. It could provide a FOV of 420 × 420 μm^2^ at a frame rate of 10 Hz at 512 × 512 pixels and a working distance of 1 mm in water. Although the resolution was reduced to 1.1 μm laterally and 12.2 μm axially due to the lower numerical aperture of the objective than that of the previous version, this new miniature TPM probe could resolve subcellular structures in neurons, as well as activity in individual dendrites and spines in the superficial layers of the cortex in vivo (Figure 10d).

The development of miniature TPM probes based on electrostatic MEMS scanning mechanisms has been summarized in Table 1. Electrostatic comb-drive-actuated MEMS scanners have been demonstrated to possess great potential for two-photon microscopic brain imaging due to their fast scan rate, high resolution, and low power consumption. However, the relatively high voltages applied may present a safety concern for intra-vital operations. In addition, the electrostatic scanners have strong nonlinearities, limiting the MEMS displacement. Compared to PZT tube-based fiber scanners, the size of TPM probes based on electrostatic MEMS could not easily be less than 2.0 mm due to the low fill factor of the MEMS mirror, which could limit its application for in vivo microendoscopic TPM imaging.

### 3.2. Electromagnetic MEMS-Based TPM

Electromagnetic actuation utilizes the Lorentz force to drive structures, which can provide a large force under low voltages. MEMS mirrors can be easily implemented by either moving magnets or coils [66]. For moving magnets, either sputtered magnetic films or bulk magnets can be used on the mirror plate, while separate coils are used to drive it. For moving coils, coils are fabricated on the mirror plate, while external magnets are required, and driving is provided by injecting electrical currents into coils. High-performance 2D electromagnetic MEMS mirrors have been developed by different groups. For example, Yalcinkay et al. designed a 2D mirror for high-resolution displaying [67] in which multi-turn spiral coils were fabricated on an outer frame for driving and the mirror plate was 1.5 mm in diameter. Electromagnetic MEMS mirrors have been implemented in miniature imaging systems, e.g., miniature OCT probes [68,69], confocal microscopy imaging [70], and LiDAR [71,72].

One electromagnetic MEMS mirror was incorporated into a TPM [73]. A gimbaled torsional stainless-steel MEMS scanner was fabricated by electrical discharge machining, offering a large inner mirror plate of 4.8 × 6 mm^2^ within a total size of 20 × 20 mm^2^. The maximum optical scan angle reached 20.6° at 112 Hz under a power of 200 mW for the slow axis, whereas the maximum optical scan angle reached 26.6° at a resonance frequency of 1268 Hz under a power of 400 mW. A bench-top TPM enabled by this MEMS mirror was used for the fluorescence imaging of a cultured cancer cell line of COLO 205. A FOV of 200 × 400 μm^2^ and a lateral resolution of 1 μm were achieved for the fluorescence TPM with an imaging rate of 10 fps. Compared with electrostatic actuation, magnetic actuation benefits from its larger scanning angle and smaller driving voltage. However, magnetic field interference may hinder its application in some situations. In addition, the relatively large size limits its implementation in mTPM applications because the overall size of the microendoscope is not adequately small to perform repetitive imaging in vivo for freely moving animals.

### 3.3. Electrothermal MEMS-Based TPM

Electrothermal actuators convert electrical energy to mechanical work through thermal expansion resulting from localized Joule heating. They may comprise a single material or multiple materials [74], which correspond to different actuation mechanisms. A thermal bimorph is a cantilever beam that consists of two or more layers of materials with different coefficients of thermal expansion (CTE). Electrothermal bimorph MEMS mirrors can generate large forces and displacements at low voltages (e.g., less than 10 V), which is ideal for biomedical applications, and they have been implemented in miniature imaging systems, e.g., miniature OCT probes [75], optical scanning [76], and TPM probes [28,77].

Buser et al. reported a 2D MEMS mirror for medical applications using Al/Si as the bimorph material and achieved 8° mechanical tilt with power of less than 180 mW [78]. Singh et al. developed a MEMS mirror based on Al/Si bimorph actuators and achieved scanning angles up to 17° with voltages less than 2 V and power less than 50 mW [79]. Compared to Al/Si materials, Al and SiO_2_ have a larger CTE difference, so Al/SiO_2_ bimorph electrothermal MEMS mirrors are expected to have a larger scan range, and several Al/SiO_2_ bimorph electrothermal MEMS mirrors have been developed [75,77,80,81,82]. In 2006, Fu et al. first applied an electrothermal bimorph 1D MEMS mirror in endoscopic TPM imaging, where the packaged probe was 3 mm in diameter [28], as shown in Figure 11a. The mirror plate was 1 × 1 mm^2^ and the maximum scan angle reached 17° at 33 Vdc (Figure 11b). Images in vitro of a rat’s esophagus tissue were acquired with the lateral and axial resolutions around 1 μm and 6 μm, respectively (Figure 11c). However, this mirror had some problems, including hysteresis of the heat response, large initial tilt, and a non-stationary center of rotation, resulting in a slow response time and low image quality [80]. To address these problems, the same team developed several generations of electrothermal MEMS designs until two-axis electrothermal MEMS mirrors based on an S-shaped inverted-series-connected (ISC) thermal bimorph actuator structure were developed [77,81,82], as shown in Figure 11d, which further increased the fill factor, eliminated both tip-tilt and lateral shift at the end of the folded beam by employing symmetric actuators, and improved the thermal response speed through the removal of the thick, rigid Si beams in the actuator. In 2020, Mehidine et al. designed a homemade TPM imaging miniature probe based on an electrothermal MEMS mirror with two-level-ladder double-S-shaped electrothermal bimorph actuators (Figure 12a) [77]. The probe had a total outer diameter of 4 mm (Figure 12c), generated a large optical scan angle of 24° driven by 4 V voltage, and achieved a FOV of 450 µm with 2 fps. This miniature imaging probe may be coupled to a two-photon microendoscope oriented towards clinical use, although it may not be quite ready yet.

A chevron actuator is another type of in-plane electrothermal actuator that relies on the total amount of thermal expansion in the structure to produce displacement in one linear direction. In 2014, Chen et al. presented a three-axis thermomechanical actuator-based endoscopic scanner for ex vivo two-photon images [21], which used one electrothermal chevron actuator to achieve in-plane x-scanning and another chevron actuator to achieve depth scanning, while a third actuator was used to drive fiber scanning to achieve y-scanning. This scanner can scan at 3 kHz × 100 Hz × 30 Hz along three axes throughout a 125 × 125 × 100 µm^3^ volume, with lateral and axial resolutions of 1 μm and 8 μm, respectively. It successfully demonstrated 3D two-photon images of fluorescent beads, but the relatively large size and limited FOV have constrained the in vivo clinical applications until now.

Compared with electrostatic or magnetic actuations, the advantages of electrothermal actuator-based MEMS scanners include large scanning angles and displacements, low actuation voltage, a high fill factor, easy fabrication, and low costs. They do not involve electrostatic or magnetic fields for operation; therefore, these devices are suitable for biomedical applications. One drawback of the electrothermal MEMS mirror-based TPM is the slow scanning speed due to the thermal response of bimorph materials, i.e., less than 5 frames per second. Fortunately, new bimorph material pairs with Cu and W have been developed [83,84] to further enhance the performance of electrothermal MEMS mirrors (Figure 13a). Cu/W bimorphs not only increase the stiffness but also speed up the thermal response time and improve reliability. For example, a MEMS mirror based on ISC bimorph actuators (Figure 13b) achieved a maximum scan angle of 47.5° at a resonance frequency of 2.7 kHz (tip-tilt mode). It had survived 200 billion cycles of large angular scanning. Electrothermal actuator-based MEMS scanners show great potential for transverse TPM imaging in vivo.

## 4. Z-Scan TPM

In the previous decade, a variety of miniature laser-scanning fiber-coupled microscopes for two-photon excitation imaging were developed. However, most of these devices suffered from a lack of active axial scanning, which greatly limited their application deep into the tissue. In order to fully study the spatially complex neuronal interconnections or identify lesions in organ tissue, an axial-scanning mechanism is required to add an additional third degree-of-freedom for 3D imaging with a miniaturized TPM. Moving the objective or sample is the most straightforward method to achieve axial scanning, as shown in Figure 14a, which has been usually implemented using a piezoelectric motor for miniaturized microscopes. For example, Piyawattanametha et al. integrated a DC micromotor into a two-photon fluorescence microscope probe to observe the neocortical microvasculature with a depth scan of 270 μm in adult mice anesthetized in 2009 [23]. One advantage of this method is that it can provide stable optical properties and a corresponding constant axial resolution, but it is not easy to integrate a motor into a miniaturized TMP for medical imaging in vivo due to its huge size. Another method is wave-front modulation or defocusing (Figure 14b). Devices such as deformable mirrors [76,85,86,87], spatial light modulators [88,89], or variable-focus lenses [90,91] have been deployed to change the focal plane to achieve depth scanning [92]. Remote focusing is another general method for axial scanning. As shown in Figure 14c, the objective near the sample is stationary and axial scanning is implemented by moving a mirror up or down with regard to the other fixed objective [93]. MEMS-based remote focusing has also been successfully employed in a two-photon imaging system [94,95]. Some recently emerging axial scanning techniques for mTPM are summarized as follows.

### 4.1. MEMS Actuated Tunable Microlens Scanner

MEMS technology can be employed to fabricate miniature microlenses with tunable focal lengths in the order of hundreds of micrometers or even millimeters in order to replace the objective for imaging deep into the sample. MEMS microlenses have been demonstrated with electrostatic, magnetic, and piezoelectric actuation. In 2004, Kwon et al. reported an electrostatic vertical-comb-drive two vertically cascaded polymer microlens scanner for a confocal microscope that achieved an axial resolution of 37 µm [96], where the loaded lens was a 400-μm-diameter polymer lens. In 2015, Michael et al. reported a piezoelectrically driven lens scanner with a 600-μm-diameter glass microlens weighing 320 μg but with a maximum displacement of only 22 µm at 60 kV/cm [97]. Thus, these existing MEMS lens scanners all encountered one or more issues such as a small travel range, high driving voltage, low stiffness, low lens quality, or small clear aperture, which limited their applications in biomedical depth scan imaging.

Electrothermal actuators are known for achieving large displacement at a low voltage and small area. In 2010, Wu et al. demonstrated an 880-µm tunable-range microlens scanner (clear aperture: 0.5 mm) based on a lateral-shift-free large-vertical-displacement (LSF-LVD) electrothermal bimorph actuator for endoscopic imaging [98], where a glass rod lens with a diameter of 1 mm was loaded, and the resultant resonance frequency was 79 Hz. In 2014, Liu et al. developed an electrothermal MEMS lens scanner (clear aperture: 2 mm) based on an LSF-LVD actuator, loaded with a plano-convex glass lens with a larger diameter of 2.4 mm (Figure 15a,b) and achieved a maximum travel range of 400 μm [99], but the resonance frequency of the scanner was only approximately 24 Hz due to its low stiffness (0.23 N/m); thus, it can be applied only in well-controlled conditions. The assembled probe with a diameter of 7 mm was verified by cooperating with a confocal scanning system to acquire confocal images with an axial resolution of 7.0 μm. Recently, Zhou et al. [100] presented one type of unique electrothermal microstage based on a single serpentine inverted-series-connected (ISC) electrothermal bimorph actuator, which had a compromised large travel range and high stiffness, so that a high-optical-quality 2.4-mm-diameter aspheric glass lens with a relatively large weight (~8 mg) could be loaded without sacrificing the axial travel range, as shown in Figure 15c,d. The stiffness of the microstage was 6.2 N/m, and axial resolution reached 2.2 µm. The resonant frequency of the MEMS microstage reached 140 Hz, and the tunable range was over 200 µm at only 10.5 V, which is acceptable for microendoscopic imaging.

### 4.2. Remote Focusing

Remote focusing in TPMs is carried out by moving a mirror to change the wavefronts and focal length of the scanning system. In 2012, Edward et al. [101] reported a TPM system for 3D two-photon imaging. The system had a lateral scan unit (LSU) comprising a pair of orthogonally mounted galvanometer mirrors for lateral scanning and an axial scan unit (ASU) including a lens and a reflective mirror for axial scanning, as shown in Figure 16a. The high-speed axial scanning with a kHz scan rate was achieved by moving a lightweight mirror M, instead of the objective, without introducing significant spherical aberration. The movement of mirror M affected the wavefront emerging at the entrance of the objective lens, as shown in Figure 16b. The two-photon imaging system was simulated to obtain in vivo images of neurons approximately 200–300 um below the cortical surface in cortical slabs with a resolution at the subcellular level without aberration (Figure 16c,d). A compact MEMS scanner with a large out-of-plane translational stroke for axial scanning in a multi-photon microscope was presented in 2017 by Li et al. [95]. The system included a bi-axial torsion MEMS mirror and an out-of-plane translational MEMS scanner performing lateral and axial scanning, respectively. The axial scanning mirror had a diameter of 0.8 mm, actuated by four electrostatic comb drives, and had vertical out-of-plane translational motion, resulting in the axial displacement of the focus. The test results on mouse colonic epithelium ex vivo showed that fluorescence images with a 3D volume of 270 × 270 × 200 μm^3^ and a frame rate of 5~10 Hz were enabled in a remote-scan, multi-photon fluorescence imaging system. In 2021, Birla et al. designed an electrothermal vertical scanning mirror with a honeycomb structure as a remote moving mirror for fast axial scanning [102]. In an electrothermal actuated MEMS mirror for z-axis motion, a thick Si layer beneath the mirror surface is usually used to increase the rigidity and decrease the bending of the MEMS mirror due to residual stress. However, the additional Si layer also increases the inertia and thermal mass of the mirror, resulting in a reduced vibration frequency and prolonged thermal response time. A honeycomb-shaped support structure beneath the mirror replacing the Si layer increased the rigidity and reduced the bowing of the mirror without additional inertia or thermal mass, as shown in Figure 17a,b. This electrothermal actuated MEMS scanner based on a bench-top TPM was used to acquire 3D images of pollen grains with a depth range of 60 um at a frame rate of 5 Hz. The imaging performance (Figure 17d,f) was comparable to that obtained by moving specimens relative to the objective, as shown in Figure 17c,e. For the remote-focusing method, the additional mirror and scanning structure may increase the overall volume and mass, resulting in most axial MEMS scanners based on a bench-top TPM, which limits its biomedical imaging application in vivo.

### 4.3. Liquid Lens

Electrically tunable lens (ETL) or electrowetting tunable lens (EWTL) technology is an ideal candidate for axial scanning that has been packaged into TPM probes for the neuronal imaging study. ETLs are well suited for microscopic applications because they allow fast changes in the focal length by varying the applied current while maintaining a large aperture size [103]. In 2021, Zong et al. developed a miniature two-photon microscope integrating a fast ETL as an axial scanning mechanism [35], as shown in Figure 18a,b. The TPM probe was 16 × 9 × 30 mm^3^ in size and had a weight of 4.2 g. The scanner, mounted on the head of a freely moving mouse, could obtain 3D imaging (Figure 18d,e) over a volume of 420 × 420 × 180 μm^3^ at a frame rate of 10 Hz and a working distance of 1 mm in water when a current ranging from −200 mA to 200 mA was provided to the ETL to change the focal length. Although the resolution was reduced to 1.1 μm laterally and 12.2 μm axially due to the lower numerical aperture of the objective compared to that in the previous version, this new miniature TPM probe has successfully resolved subcellular structures in neurons, as well as activity in individual dendrites and spines in the superficial layers of the cortex of a freely moving mouse.

Electrowetting tunable lens (EWTL) technology allows for variable focus by using an applied electric field to change the curvature between two immiscible fluids. The advantages of EWTLs include a high scan speed (~50 Hz over the full focal range), long-term repeatability, miniaturization to mm-scales, insusceptibility to motion and orientation, and low power requirements [104]. In 2018, Ozbay et al. presented a miniature head-mounted two-photon fiber-coupled microscope (TP-FCM) for neuronal imaging with active axial focusing enabled using a miniature electrowetting lens [88], as shown in Figure 19A,B. The EWTL had a clear aperture of 2.4 mm diameter and an outer diameter of 7.8 mm, with the tuning focal length driven by a voltage between 25 and 60 V_RMS_ at 1 kHz. Here, 3D two-photon images of neuronal structures were shown and neuronal activities from GCaMP6s fluorescence from multiple focal planes in a freely moving mouse were recorded, as shown in Figure 19D,E. Additionally, dynamic control of the axial scanning of the electrowetting lens allows the tilting of the focal plane, enabling neurons in multiple depths to be imaged in a single plane, as illustrated in Figure 19F. The TPM headpiece had a weight of only 2.5 g and yielded a working distance of 450 μm with an additional 180 μm of active axial focusing. The objective NA was 0.45, with lateral and axial resolutions of 1.8 μm and 10 μm, respectively, and a FOV of 240 μm in diameter. Fluorescence imaging tests on the same mouse over 17 days demonstrated the stability of the TP-FCM on a baseplate. Therefore, ETL and EWTL technologies have been successfully demonstrated to be applied in neuroscience research, while efforts are still needed to reduce their size for packing into a smaller probe for microendoscopic applications.

### 4.4. Deformable MEMS

An ultrafast non-mechanical axial scanning method for two-photon excitation microscopy (TPEM) based on binary holography using a digital micromirror device (DMD) was presented by Cheng et al. in 2016 [105]. The principle of the DMD scanner is to shape the wavefront of an input beam into a spherical wavefront via the DMD, where binary holograms of spherical wavefronts of increasing/decreasing radii are programmed. As shown in Figure 20a,b, the radius of the spherical wavefront directly determines the location of the focal point to realize axial scanning. The DMD scanner achieved an axial scanning range of approximately 180 μm and a scanning resolution of approximately 270 nm with a scanning rate of 4.2 kHz. Compared to a precision z-stage image of a pollen sample, the DMD scanner integrated into a custom-built TPE microscope operating at 60 frames per second could generate images of equal quality through the scanning range FOV with 200 × 200 μm^2^ and 512 × 512 pixels, as illustrated in Figure 20c.

Until now, only ETL or EWTL integrated into a miniaturized TPM probe mounted on a freely moving animal have been realized to perform axial scanning and obtain 3D two-photon images of neurons in vivo. Other axial scanning techniques and components have verified the feasibility of z-axis scanning imaging based on a bench-top or custom-built TPM. Of course, no matter which technique is applied to two-photon microendoscopic imaging, it is necessary to further reduce the size of the probe and improve the scanning rate.

## 5. Discussion and Summary

In this article, we discussed miniature TPM based on various MEMS micromirrors, including electrostatic, electrothermal, and electromagnetic micromirrors. Among them, electrostatic MEMS mirror-based TPM can achieve a fast scan rate, high resolution, and low power consumption, but it has a low fill factor of the MEMS mirror and requires high voltage. In principle, the electromagnetic MEMS mirror-based TPM has a relatively high generated force and can achieve large scanning angles. The main drawbacks are high-power dissipation as well as complex fabrication, and difficulty in miniaturization. Electrothermal MEMS mirror-based TPM can achieve large scanning angles and displacement with low driving voltage, low-cost fabrication, and compact devices but slow response time due to large currents. Overall, electrostatic and electrothermal actuation-based MEMS mirrors are more promising for miniature TPM. We also discussed miniature TPM with a z-axis scan. MEMS-actuated tunable microlenses exhibit small size and large travel range but low stiffness to load high-quality lenses. TPM with remote-focusing can acquire images of neurons; however, the complicated optical structure makes excessive system size. Liquid lenses, such as ETL or EWTL, exhibit high performance for TPM with dynamic axial scanning on a moving mouse. Deformable MEMS mirror actuated by digital circuit has high scan rates and submicro axial resolution, especially for high scan display. Overall, miniature TPM with a liquid lens is more promising for 3D imaging.

In summary, miniaturized two-photon microscopy (TPM) provides a powerful means for studying neuronal activities in real-time on freely moving small animals, as well as diagnosing diseases in internal organs without biopsy. In this paper, we have introduced the principles and designs of various miniaturized TPM systems. More importantly, we have reviewed the progress and challenges of miniaturized TPMs. For two-photon microendoscopic applications, fiber tip scanners, with flexibility and a small diameter, can be directly inserted into the biopsy channel of a conventional endoscope for transverse imaging. Nonetheless, large efforts are still needed to further improve the scan speed and image uniformity and decrease the driving voltage. In addition, axial scanning capability must be realized in a small form factor for 3D imaging for clinical diagnosis or surgical guidance. For two-photon brain imaging, electrostatically actuated MEMS scanners plus ETL or EWTL, packaged into a compact headpiece mountable on an awake mouse, seem to be the most promising to perform long-term recurring two-photon 3D imaging.

In the future, with the advances in MEMS microfabrication and semiconductor laser technology, all of the optical components can be integrated, so a smaller size, wider axial scan range, larger FOV, higher scanning rate, and higher resolution can be achieved for in vivo microendoscopic imaging or brain neuronal study. Furthermore, other aspects, including the ease of sterilization, electrical safety, stability, reliability, and a lifetime of the devices, and the cost of fabrication, must be considered for clinical applications.

## Figures and Tables

**Figure 1 micromachines-14-00470-f001:**
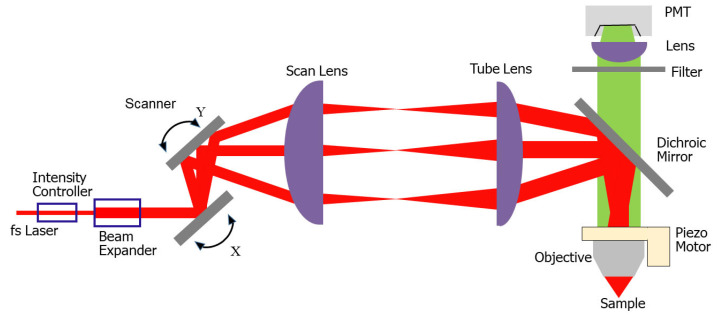
Simplified schematic diagram of TPM.

**Figure 3 micromachines-14-00470-f003:**
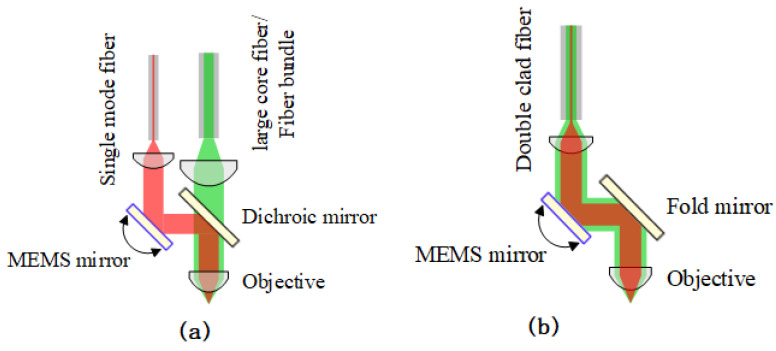
Schematic of fiber-based TPM. (**a**) Two-fiber configuration. (**b**) Single-fiber configuration.

**Figure 4 micromachines-14-00470-f004:**
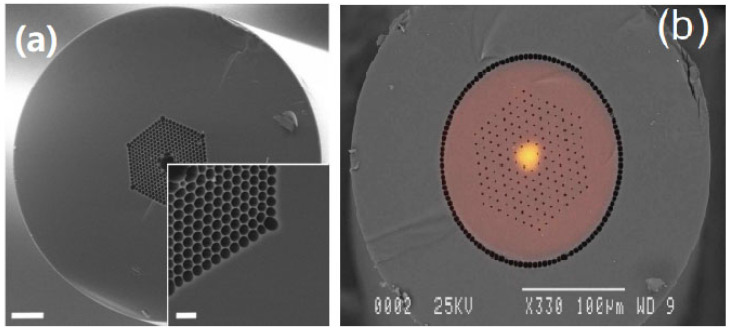
(**a**) SEM micrographs of the air-core PCF and cladding structure. Reproduced with permission from [26]. Copyright 2008, OSA. (**b**) A far-field output pattern from a double-clad PCF at wavelength 800 nm overlaid on an SEM image. Reproduced with permission from [28]. Copyright 2006, OSA.

**Figure 5 micromachines-14-00470-f005:**
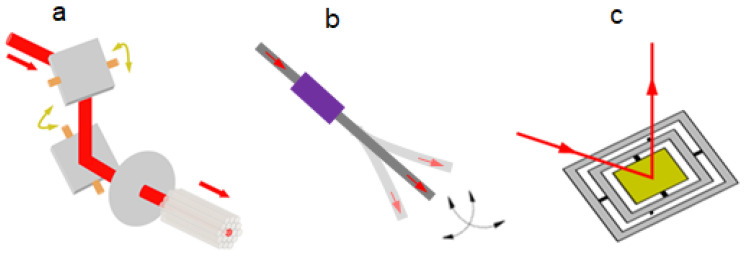
In-plane scanning mechanisms. (**a**) Proximal scanning. (**b**) Distal fiber tip scanning. (**c**) MEMS mirror scanning.

**Figure 6 micromachines-14-00470-f006:**
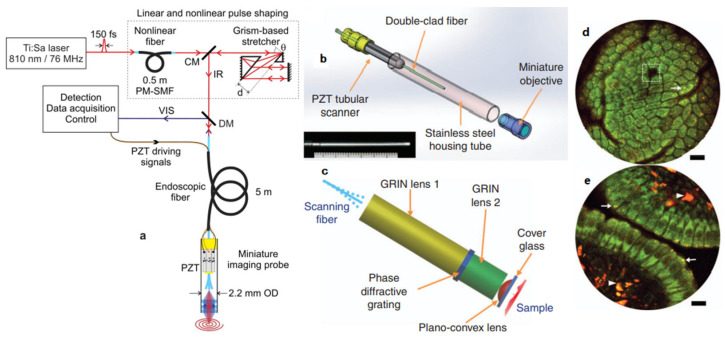
(**a**) Scheme of the PZT-based fiber scanning probe with 2.2 mm outer diameter. Reproduced with permission from [44]. Copyright 2015, Springer Nature. (**b**) Photograph of the microendoscopic head with an outer diameter of 2.1 mm and a length of 35 mm. (**c**) Illustration of the customized miniature objective. (**d**,**e**) Microendoscopy 2PF and SHG label-free structural imaging of the mucosa of mouse small intestine in vivo, respectively. Reproduced with permission from [32]. Copyright 2017, Springer Nature.

**Figure 7 micromachines-14-00470-f007:**
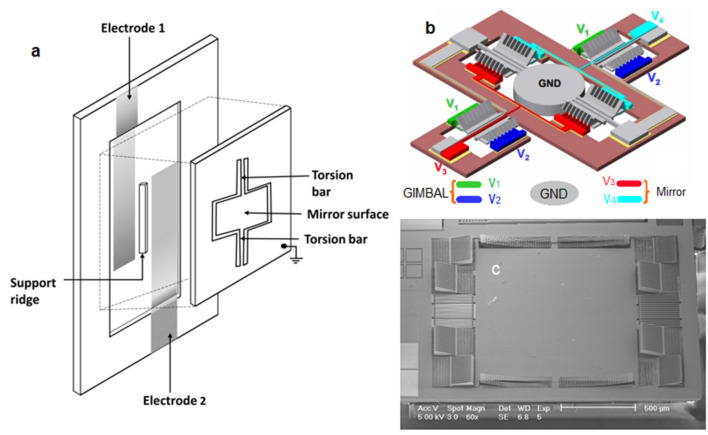
MEMS mirrors are based on electrostatic actuation. (**a**) Expanded view of parallel plate mirror structure. (**b**) Schematic of a MEMS two-axis optical scanner based on angled vertical comb actuators. Reproduced with permission from [57]. Copyright 2007, OSA. (**c**) SEM of a MEMS single-axis optical scanner based on curled-hinge vertical comb actuators. Reproduced with permission from [58]. Copyright 2003, IEEE.

**Figure 8 micromachines-14-00470-f008:**
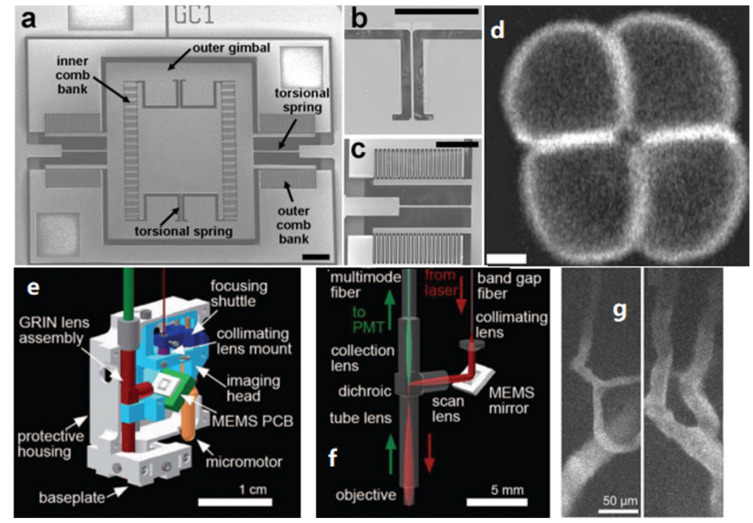
(**a**) A 2D electrostatic MEMS mirror based on vertical comb actuators. (**b**) Inner axis torsional spring. (**c**) Outer axis comb bank. (**d**) Two-photon fluorescence image of pollen grains. Scale bars are 5 μm. Reproduced with permission from [64]. Copyright 2006, OSA. (**e**) Computer-aided-design model of the microscope, in a cut-away view. (**f**) Optical pathways, red excited laser, and green fluorescence. (**g**) Images of neocortical capillaries averaged over eight frames acquired over 2 s at 4 Hz. Reproduced with permission from [23]. Copyright 2009, OSA.

**Figure 9 micromachines-14-00470-f009:**
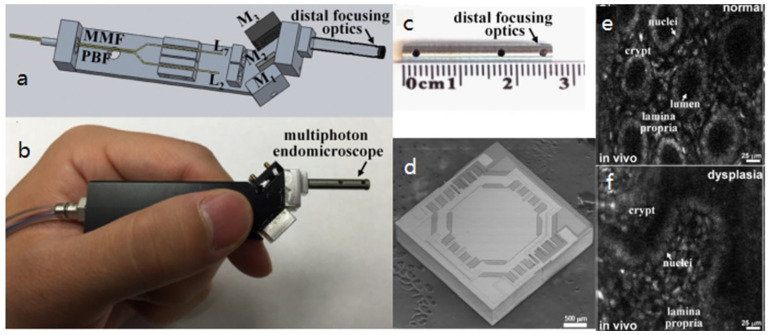
An endoscopic MPM probe based on an electrostatic MEMS mirror. (**a**) CAD drawing. (**b**) Package MPM probe. (**c**) Distal focusing optics. (**d**) SEM of 1.8-mm-diameter MEMS mirror on a 3 × 3 mm^2^ die. (**e**,**f**) Two-photon excited fluorescence images of normal colonic mucosa collected in vivo at 5 frames/s. Reproduced with permission from [25]. Copyright 2015, OSA.

**Figure 10 micromachines-14-00470-f010:**
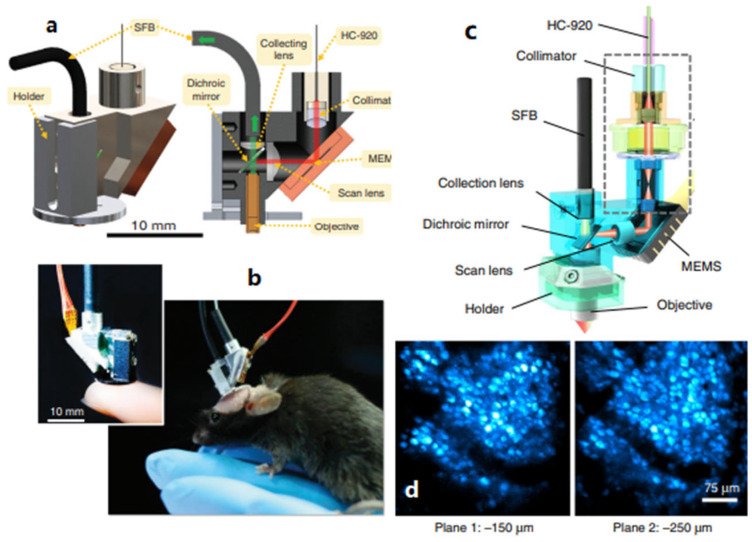
(**a**) Schematic of a fast high-resolution miniaturized (FHIRM) TPM. (**b**) Photographs of a FHIRM-TPM on a fingertip and mounted to the head of a mouse. Reproduced with permission from [27]. Copyright 2017, Springer Nature. (**c**) Schematic of the headpiece of the FHIRM-TPM 2.0. (**d**) Images of two planes at −150 and −250 μm of mPFC neurons expressing GCaMP6s in cortical layer. Reproduced with permission from [35]. Copyright 2021, Springer Nature.

**Figure 11 micromachines-14-00470-f011:**
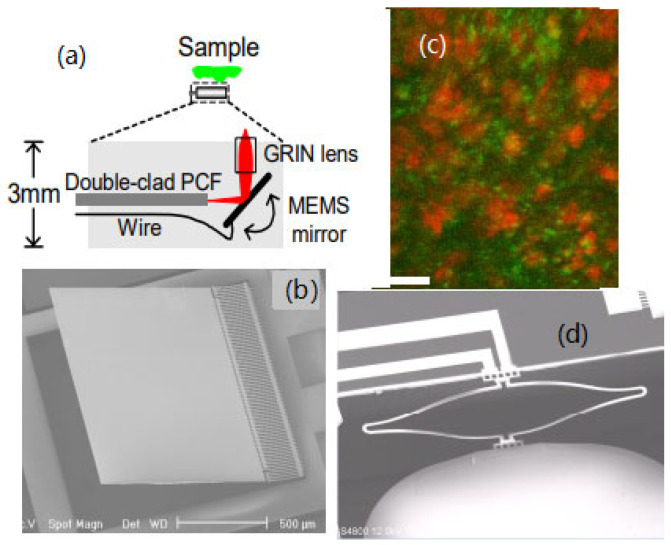
An endoscopic TPM probe based on a 1D electrothermal bimorph MEMS mirror. (**a**) Schematic diagram of the nonlinear optical endoscope probe. (**b**) An SEM image of the 1D electrothermal MEMS mirror. (**c**) Two-photon fluorescence (red) and SHG (green) visualize cell nuclei and connective tissue, respectively. Scale bar represents 20 μm. Reproduced with permission from [28]. Copyright 2006, OSA. (**d**) SEM of an ISC actuator. Reproduced with permission from [82]. Copyright 2017, MDPI AG.

**Figure 12 micromachines-14-00470-f012:**
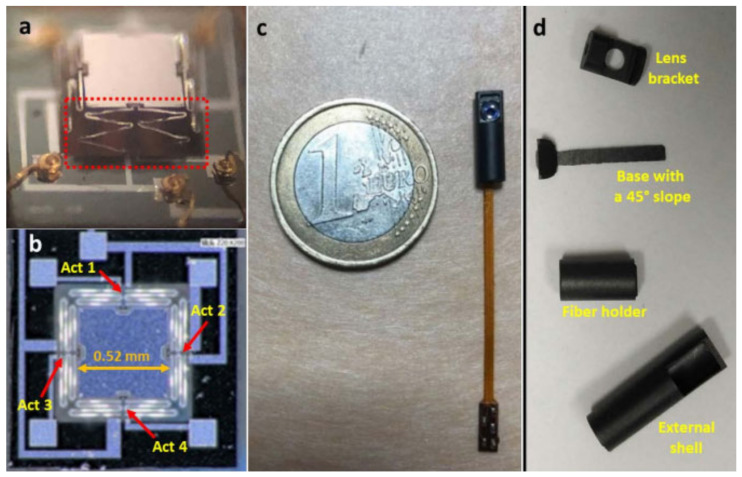
(**a**) An optical image of a MEMS mirror showing one of its two-level-ladder, double-S-shaped actuators (red square). (**b**) SEM image showing the four actuators with the mirror plate. (**c**) Assembled probe. (**d**) Probe’s mechanical components. Reproduced with permission from [77]. Copyright 2020, MDPI AG.

**Figure 13 micromachines-14-00470-f013:**
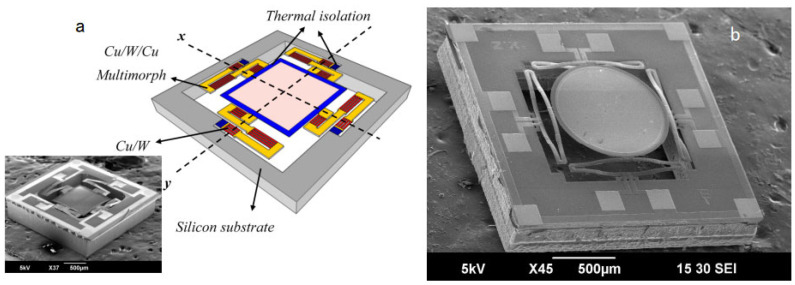
MEMS mirrors using Cu/W bimorphs. (**a**) SEM of an LSF actuator-based MEMS mirror. Reproduced with permission from [80]. Copyright 2015, MDPI AG. (**b**) SEM of an ISC actuator-based MEMS mirror. Reproduced with permission from [81]. Copyright 2019, MDPI AG.

**Figure 14 micromachines-14-00470-f014:**
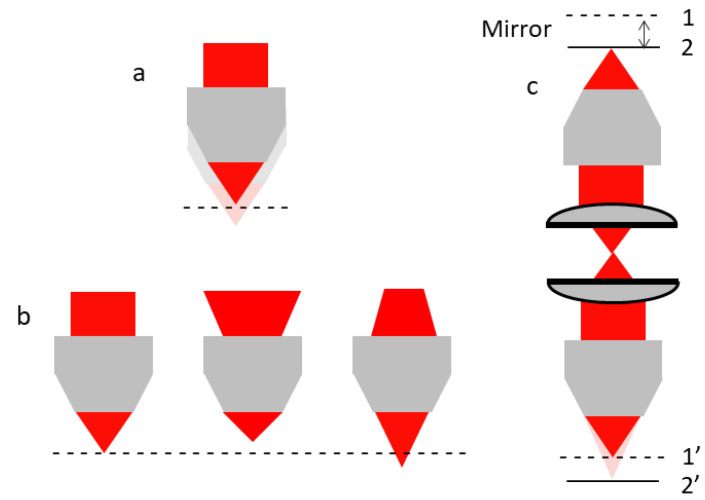
Axial scanning mechanisms. (**a**) Objective moving. The objective is actuated to move along the axial direction. (**b**) Wave-front modulation or defocusing to change the focal plane. (**c**) Remote focusing.

**Figure 15 micromachines-14-00470-f015:**
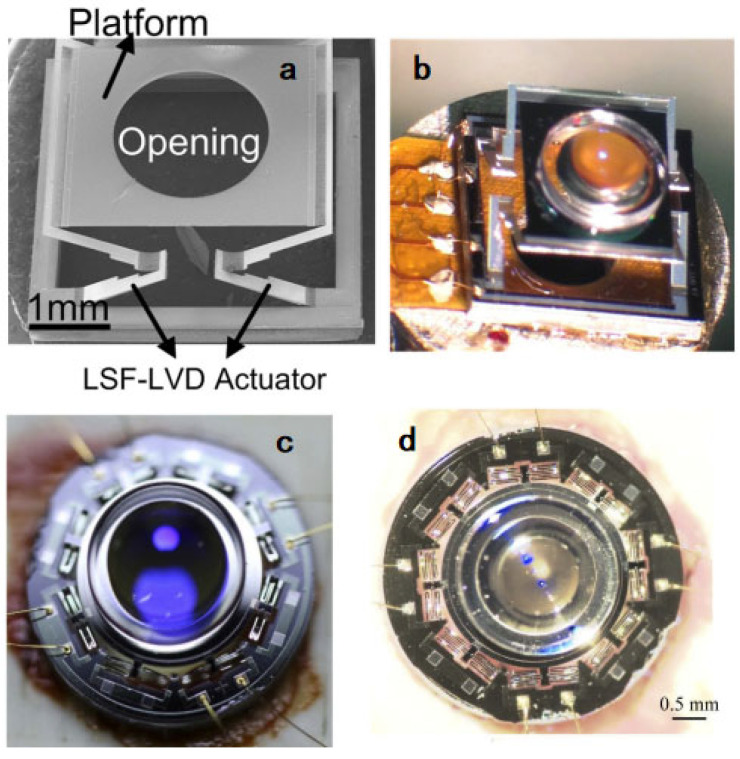
(**a**) SEM of an LSF-LSD actuated lens scanner. (**b**) Photo of an assembled tunable lens. Reproduced with permission from [99]. Copyright 2015, Elsevier. (**c**) Photo of an ISC bimorph actuator MEMS lens scanner facing upward. (**d**) Photo facing downward. Reproduced with permission from [100]. Copyright 2020, OSA.

**Figure 16 micromachines-14-00470-f016:**
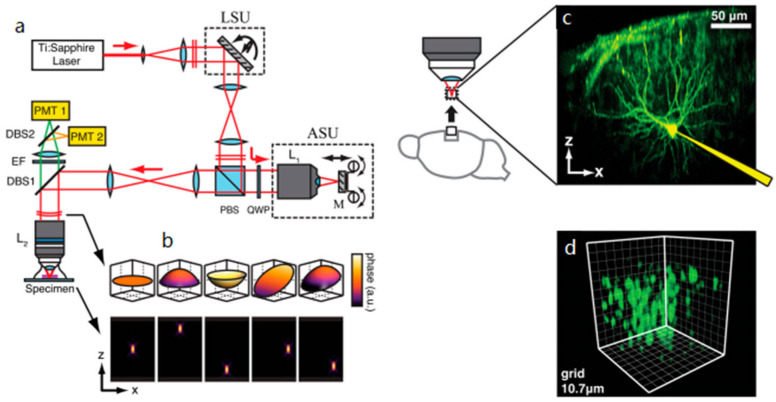
Schematic of the fast-focusing two-photon imaging system. Lateral scanning was carried out by two galvanometers in the lateral scan unit (LSU) and z-scanning by the mirror M in the axial scan unit (ASU). (**a**) Schematic of the fast-focusing two-photon imaging system. (**b**) Theoretical wavefronts and intensity distributions produced for various settings of the LSU and ASU. (**c**) Imaging was performed in cortical slabs to emulate the use of this method in the in vivo setting. (**d**) Three-dimensional rendering of neuronal population bolus-loaded with OGB1-AM. Reproduced with permission from [101]. Copyright 2011, PNAS.

**Figure 17 micromachines-14-00470-f017:**
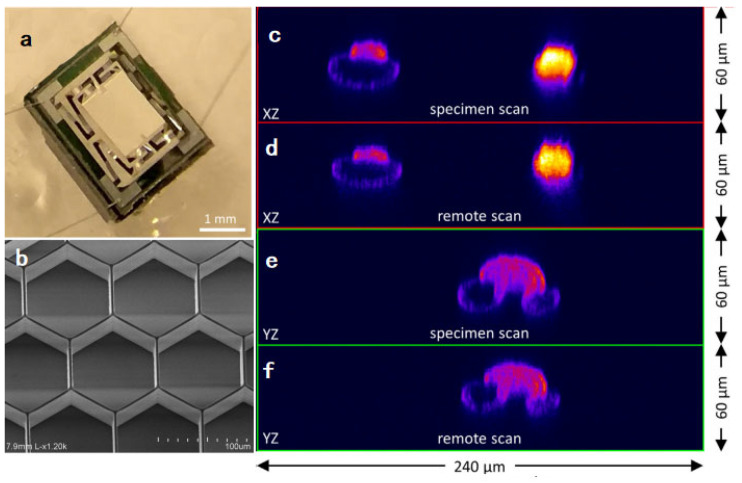
Remote axial scan two-photon microscopy system. (**a**) Image of the electrothermal scanner in which the mirror is located at the center, supported by four legs each having inner and outer folded U-shaped actuators. (**b**) SEM image of backside of a scanning mirror supported by a honeycomb-shaped structure. (**c**,**e**) Images of pollen grains obtained by moving specimen. (**d**,**f**) Images were obtained by driving the electrothermal scanner along the axial direction. Reproduced with permission from [102]. Copyright 2021, Elsevier.

**Figure 18 micromachines-14-00470-f018:**
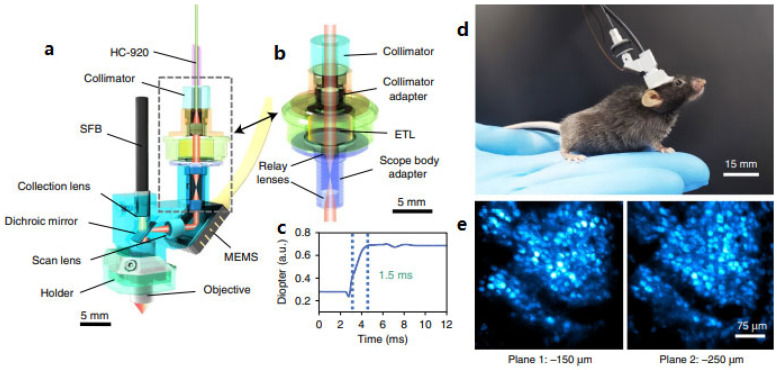
(**a**) Schematic of the headpiece of a miniature TPM. (**b**) Z−scan module. (**c**) Step response of ETL diopter change. (**d**) Photograph of a mouse wearing the mTPM headpiece. (**e**) Two planes at −150 and −250 μm were alternately imaged for 400 s at 5 Hz. Reproduced with permission from [35]. Copyright 2021, Springer Nature.

**Figure 19 micromachines-14-00470-f019:**
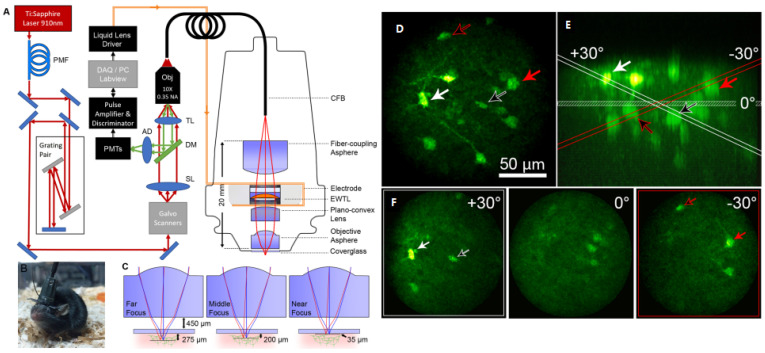
Two-photon fiber-coupled microscope system and images. (**A**) Setup of TPFCM. (**B**) Photo of a normally behaving mouse with TP−FCM attached. (**C**) Zemax simulation of 2P−FCM axial-scanning using specified EWTL lens with a working distance of 450 μm. (**D**) Maximum intensity projection of a thick coronal section of the mouse brain, expressing GCaMP6s in neurons. (**E**) Side (XZ) projection of the image volume in d. (**F**) Images showing the tilted−field scans indicating the same cell bodies that are shown to intersect with the red or white planes in e. Reproduced with permission from [91]. Copyright 2018, Springer Nature.

**Figure 20 micromachines-14-00470-f020:**
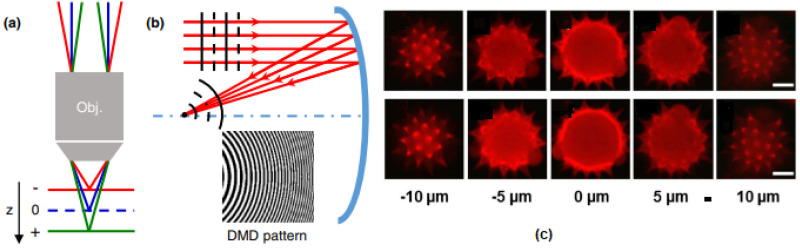
Axial scanning by DMD scanner and cross−sectional images of a pollen grain at various depths. (**a**) A convergent (red)/divergent (green) beam scans the focal point toward/away from the objective lens. (**b**) DMD scanner functions as a concave mirror with a positive focal length. (**c**) Top: optical cross−sections obtained by a precision z stage; Bottom: optical cross-sections by the DMD scanner; scale bar = 10 μm. Reproduced with permission from [89]. Copyright 2016, OSA.

**Table 1 micromachines-14-00470-t001:** Summary of miniature TPM probes based on electrostatic comb-drive-actuated MEMS mirrors.

	TPM Probe					MEMS Mirror				
	Size (mm^3^)	FOV (µm × µm)	Frame Rate (Hz)	Resolution (µm)	Working Distance (µm)	Mirror Size (mm^2^)	Chip Size (mm^2^)	Optical Angle (°)	Drive Voltage (V)	Resonance Frequency (kHz)
2006 [64]	-	250 × 90	-	~1	35~47	0.75 × 0.75	3.2 × 3.0	16	45~58	inner 3.52 outer 1.02
2008 [26]	10 × 15 × 40	Φ: 310 µm	10	lateral 1.6 axial 16.4	210	0.5 × 0.5	-	inner ±10 outer ±10.5	80	1.54, 2.73
2008 [33]	Φ: 10 long: 140	128 × 128 pixels	10	-	-	Φ: 2 mm	-	14	-	1.26, 0.784
2009 [24]	Φ: 10 mm long: 140	720 × 720 pixels	0.25	2	210	Φ: 2 mm	3.3 × 2.6	20	90	1.26, 0.784
2009 [23]	20 × 19 × 11	295 × 100	15	lateral 1.3 axial 10.3	280	1 × 1	-	inner ±5 outer ±4.3	45	1.08, 0.56
2021 [35]	16 × 9 × 30	420 × 420	10	lateral 1.1 axial 12.2	1 mm	-	-	±4.5	-	-
2017 [27]	1 cm^3^	130 × 130	40	lateral 0.6 axial 3.4	170	Φ: 0.8 mm	9 × 9	±10	10	6
2015 [25]	Φ: 3.4 mm long: 26 mm	300 × 300	5	lateral 2 axial 9.0	60	Φ: 1.8 mm	3 × 3	±4.5	40	2.91, 0.805

## Data Availability

Not applicable.

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
