# Peer review of "MEMS Enabled Miniature Two-Photon Microscopy for Biomedical Imaging"

_micromachines, 2023, doi:10.3390/mi14020470_

Round 1

Reviewer 1 Report

Some of the figures are blurred. This is a review paper, i guess you can ask the original authors of the papers to provide high resolution figures. Thanks,

Reviewer 2 Report

Journal: Micromachines

Manuscript ID: micromachines 2205811
Type of manuscript: Review
Title: MEMS Enabled Miniature Two-Photon Microscopy

Comments:

X. Yu et al have reviewed comprehensively on the MEMS-based two-photon microscopy (TPM) for clinical imaging applications. Generally, the contents of the review are fine technically.  However, there are a few points that need to be addressed accordingly before acceptance and published by Micromachines journal. Please find the comments below:

1.     “MEMS” in the title should be spelled in complete term, not abbreviated term. Please amend accordingly. Also, the overall review and discussion focus on clinical applications, thus the term “clinical applications” could be included in the title as well. The suggested title could be “Microelectromechanical System (MEMS) Enabled Miniature Two-Photon Microscopy”.

2.     Novelty and significance of manuscript: I find that the authors did not highlight the novelty and significance of manuscript to demonstrate the worthiness of publishing this manuscript in the introduction part. The current version did not portray the significance of this review. Are there reviews pertaining to this topic? If yes, what is / are the difference(s) of this review versus others?

3.     What is the outlook or future research directions of MEMS-based TPM? It is highly suggested to include a dedicated section of this aspect, prior to the conclusion.

4.     English language and grammar problems: I suggest the authors to check the English language and grammar by native English speakers. 
